# Use of Health Services and Support Resources by Immediate Family Members Bereaved by Suicide: A Scoping Review

**DOI:** 10.3390/ijerph191610016

**Published:** 2022-08-14

**Authors:** Silje L. Kaspersen, Jorid Kalseth, Kim Stene-Larsen, Anne Reneflot

**Affiliations:** 1Department of Health Research, SINTEF Digital, Pb. 4760 Torgarden, 7465 Trondheim, Norway; 2Department of Mental Health and Suicide, Norwegian Institute of Public Health, 0456 Oslo, Norway

**Keywords:** suicide, bereavement, postvention, support, internet, help-seeking, needs

## Abstract

The knowledge on health service use, systematic follow-up, and support for families bereaved by suicide remains scarce. This scoping review includes studies from 2010 to March 2022 that investigate the follow-up and support offered by health services, peer support services, and other resources available (e.g., internet-based resources) for families bereaved by suicide. We followed the scoping review framework provided by the Johanna Briggs Institute and performed a double-blinded screening process using Covidence. Data were extracted by four researchers and a thematic analysis was performed to summarize the results. The PRISMA Extension for Scoping reviews was used for reporting results. Of 2385 studies screened by title, 190 by abstract, and 93 by full-text reading, we included 63 original articles of which 24, 29 and 10 were quantitative, qualitative, or mixed-methods studies, respectively. The review shows that we have some knowledge about the need for, and experiences with, health services and support resources for immediate family members bereaved by suicide, but a lack of knowledge about their help-seeking behaviour, patient pathways, systematic follow-up, coordination between services, and long-term outcomes. We need more longitudinal observational studies of health service use and patient trajectories for people bereaved by suicide.

## 1. Introduction

One of the world’s key public health challenges is the loss of nearly 800,000 lives to suicide each year. It has been one of the leading causes of death in young adults for decades, despite continued efforts in suicide prevention and research [1,2,3]. From the onset of the COVID-19 pandemic, concern was raised about the impact of the pandemic on suicide rates, given the profound changes that took place in people’s daily and working lives [4]. So far, cautious conclusions in the research literature from the first year of the pandemic state that it did not lead to excess mortality from suicide in most of the countries studied. However, there might be large variations between subgroups in the population, and the long-term effects have not yet been measured [5,6].

What we do know is that for each life lost to suicide, several people will be bereaved for the rest of their lives. Those who have lost a loved one to suicide are often referred to as people bereaved by suicide in the literature [7,8,9]. Traditionally it has been assumed that at least six close family members are left behind for each suicide. Some studies have estimated an average of 135 people affected by each suicide, and there is an ongoing debate on who should be looked upon as exposed and included in the bereavement numbers [8]. Pitman et al. [9] estimated that 48 to 500 million people worldwide experience suicide bereavement every year. However, only a small proportion of the suicidology literature has studied the outcomes, needs and characteristics of people bereaved by suicide [10]. Helping the bereaved to handle their grief and have a good life will benefit both the individual and society as a whole. It may also act as suicide prevention per se, since people bereaved by suicide are at a 2–3 times higher risk of suicidal behaviour, compared with the general population [11].

People bereaved by suicide may be of higher risk than those bereaved by other causes of developing complicated grief (CG) reactions and health problems [9,12,13,14]. Struggling with feelings of shame and blame, social stigma, rejection and abandonment are aspects that often occur in those who have lost someone to suicide [9,14,15]. Two Norwegian studies found that 70–80% of suicide-bereaved parents and adolescents reported that they needed medical or psychological care in relation to the bereavement experience [16,17]. Bereaved children after parental suicide are particularly vulnerable, with increased risk of depression, poor educational performance, and suicide [9,18,19,20]. A recent study of Italian suicide survivors found that 57% of the participants sought professional help after the loss [21]. An Australian study exploring experiences with use of resources that provide support to people affected by suicide-related behaviour found that those bereaved by suicide were the least likely to seek support immediately after the suicide-related behaviour, compared to those who had attempted suicide or were carers [22].

The existing literature on suicide bereavement has mainly focused on specific health outcomes and postvention [23]. Maple et al. (2018) mapped the suicide bereavement and postvention literature and characterised 443 articles in terms of their methodological design, year of publication, age and population groups, and geographic spread [24]. Bartone et al. (2019) systematically reviewed the literature on the effects of peer support for bereaved survivors, including counselling from peers, support groups and internet forums [25]. Andriessen et al. (2019) reviewed the evidence for the effectiveness of several interventions for people bereaved by suicide [26], while Lestienne et al. (2021) completed a systematic review of online resources and interventions for people bereaved by suicide [27]. However, knowledge about health service use and patient pathways, systematic follow-up and support of families bereaved by suicide remains scarce.

In this study, we describe the findings from a scoping review performed between June 2020 and March 2022, including studies from 2010 and onwards, that investigated the follow-up and support of families bereaved by suicide from health services, support groups and other resources available for support besides family and friends. While our initial focus was on health services and other forms of organised support (both within and outside traditional health services), the recent review of Lestienne et al. (2021) [27] raised our awareness that online resources may increase the availability of support for the suicide bereaved facing barriers to follow-up from health services and other organised resources. Internet-based mutual support resources such as online forums, chat rooms and online memorials should be considered in postvention strategies and policies. Hence, we extended our approach to allow for online mutual support resources. In the remainder of this article, when using the term support, we include online mutual support resources.

With an elevated focus on suicide and health service resources during the pandemic, we expected a boost in research on suicide and suicide bereavement from 2020, and the original search performed according to protocol in June 2020 [28] was updated in March 2022 to include research from the first- and mid-phase of the pandemic.

### 1.1. Objective

The objectives of this scoping review are: (1) to scope the extent and nature of the scientific literature on the support, follow-up and use of health services and other available support resources by suicide-bereaved families; (2) to give a detailed thematic overview of the findings in the literature by types of support; and (3) to identify research gaps.

### 1.2. Study Rationale

This scoping review is part of an ongoing research project called “Treatpath” led by the Norwegian Institute of Public Health in collaboration with SINTEF, the Department of Health Research (independent research institute), and Oslo University Hospital. A preliminary literature search while formulating the study proposal for Treatpath showed that there is limited knowledge on the role that health services and professionals play in the follow-up and support of people bereaved by suicide.

## 2. Methodology

We used scoping review methodology to systematically identify original articles following a pre-established search strategy and published protocol [28]. We followed the scoping review framework provided by the Joanna Briggs Institute (JBI) [29,30], which in turn is based on earlier efforts by Arksey and O’Malley [31] and Levac and colleagues [32]. The JBI approach to conducting and reporting scoping reviews is congruent with the PRISMA Extension for Scoping Reviews checklist [29,33], which was used as a guide for reporting the results [34]. The review process followed six stages: (1) identifying the research question, (2) identifying relevant studies, (3) selecting studies (4) charting the data (5) collating, summarising, and reporting the results, and (6) consulting the project team. A thorough explanation of each step is described in the protocol article [28].

## 3. Search Strategy

### 3.1. Preliminary Search in Systematic Reviews

In the preparation phase of the scoping review, we conducted a preliminary search in Medline (via PubMed) and PsycINFO (via OVID) in May 2020 in order to obtain an overview of the research field and establish relevant search terms [28].

### 3.2. Population–Concept–Context

We used the population–concept–context (PCC) framework provided by the Joanna Briggs Institute to identify relevant studies (Table 1) [29]. The main outcomes for which data were sought were health service use, follow-up, and support for immediate family members bereaved by suicide. However, studies that included both immediate family members (defined as parents, spouse/partner, siblings, and children of any age) and other bereaved people were included if most participants were within the immediate family definition.

Included studies had to meet the following criteria: (1) publication in a peer-reviewed journal, (2) original empirical study, (3) inclusion of participants bereaved by suicide, (4) study aim or outcome including follow-up, support, or health service utilisation by families bereaved by suicide, (5) publication between January 2010 and March 2022, and (6) study from a high-income country according to the World Bank.

We excluded grey literature, books/book chapters, reports or unpublished studies, and studies of bereavement after euthanasia and physician-assisted suicide. The exclusion of low-income countries was chosen to increase the probability of comparable levels of healthcare services and available support in the included literature. Since we wanted to capture “up to date” follow-up in the health services we restricted the search to publications from 2010 to March 2022.

We conducted our main search using the following databases: Medline (via Ovid), Embase (via Ovid), PsycINFO (via Ovid), CINAHL (via EBSCO host) and Web of Science. We used both subject headings (e.g., MeSH, major heading, subject terms, Keywords Plus) and free text searching. The search strategy is presented in Table 2. An example of the search strategy in Ovid (APA PsycInfo, Embase and Medline) is shown in Appendix A.

### 3.3. Search Results

There were, in total, 3835 records retrieved from CINAHL, Embase, Medline, PsycINFO, and Web of Science. After removing duplicates, the remaining records were screened by title and irrelevant records were excluded. The remaining articles were screened by abstracts and a selection of these were included for full-text reading. Full-text reading led to the further exclusion of papers due to the following reasons: not an original article, not immediate family members, participants not bereaved by suicide only, not follow-up or support, focus on events prior to suicide, etc. See the PRISMA diagram in Figure 1 for details. We supplemented the search with the CoCites citation-based search method [35] used in the systematic reviews retrieved in the preliminary search described above, but this did not provide us with any undiscovered articles.

In order to make a final quality check of our search strategy, we did an additional search of systematic reviews conducted in March 2022 in Pubmed (search period 1 January 2010–15 March 2022) and found six relevant systematic reviews on follow-up and support for the suicide bereaved [12,26,27,36,37,38]. We screened the included studies in the six reviews and found that all included studies were identified in our search. However, nine of the studies had been excluded in our screening process (title/abstract screening phase) but were reconsidered to be of relevance. Of these, one was on family-based therapy included in both Andriessen et al. (2019) [26] and Linde et al. (2012) [12], one on support groups included in Higgins et al. (2022) [37], and seven were studies on the use of internet-based resources found in Lestienne et al. (2021) [27].

### 3.4. Screening Process

The studies were imported into EndNote 20 and Covidence. EndNote 20 is a reference-management software from Clarivate Analytics used to administrate references, while Covidence is available from the Cochrane Library and was used to remove duplicates and screen titles and abstracts (Figure 1). Two pairs of team members screened titles and abstracts in a blinded voting process (standard in Covidence). The team members were unaware of what the other member had voted with regard to eligibility. When conflicting results occurred, a final decision on inclusion/exclusion was made by discussion and consensus.

All authors were involved in full-text reading. Full-text reading was completed in two sequences: (1) rapidly screening full-texts for eligibility and (2) thoroughly reading full-texts, extracting data and narratively summarizing results. Several meetings were held to discuss the data extraction process and the thematic synthesis of the results.

### 3.5. Data Extraction

A data-charting form was jointly developed in Excel by SLK and JK. All four full-text readers charted the included studies in an evidence table with the following pre-defined categories: author(s); year of publication; origin/country of origin (where the study was conducted); aim/purpose of the study; study population and sample size (if applicable); methodology/methods and key findings that relate to the scoping review question(s) (e.g., type, amount, timing and range of services/interventions, patient trajectories, patient experiences and outcomes). Data extraction was performed independently by the four researchers with dialogue meetings during the process.

### 3.6. Collating, Summarising and Reporting the Results

We followed the checklist for reporting in scoping reviews—the Preferred Reporting Items for Systematic Reviews and Meta-Analysis: extension for Scoping Reviews (PRISMA-ScR) [34]. We wanted the review to have a broad perspective on health service use and support for the suicide bereaved and chose to provide detailed information about the results of the studies. Hopefully this approach will make the review useful for clinicians in general health services and can support resource facilitators and researchers within health service research that are not necessarily familiar with suicide bereavement.

To summarise the results, we used an approach inspired by thematic synthesis [39,40]. First, information on service/support types and themes were extracted and coded. We also coded whether the study focused on one service or support type, whether the study focused on internet-based services/support types, and whether the services/support types included were postventions, i.e., service/support targeted towards the suicide bereaved. Next, based on the service/support type and themes coded, studies were grouped into a smaller number of categories. Based on the results, for services/support types we developed mutual exclusive categories, while for themes the categories were non-mutually exclusive. The reporting of results also involved providing key quantitative information including publication year, country, aim/purpose, study population and sample size, study design/methodology, and key findings. We also completed a network analysis of the publications included, using the library tool VOSviewer (see Appendix A). This scoping review did not assess the quality of or risk of bias in the included studies.

### 3.7. Consultation

We had meetings with the Treatpath research team to discuss the findings of the review and how to interpret and report them. The researchers in the team are experts on suicide research in Norway. We also plan to present the results to The Norwegian Organization for the Suicide Bereaved and use their channels for communication of the results to the suicide bereaved.

### 3.8. Patient and Public Involvement

Patients and the public were not consulted as part of this scoping review as it was not appropriate or applicable. However, The National Association of Suicide Bereaved in Norway (LEVE) has a representative in the Treatpath project’s advisory board and has been involved in the application process for project funding and research group meetings.

### 3.9. Ethics

The methods of a scoping review aim only to scope and map previously published research, and ethical approval is generally not considered necessary. The “Treatpath” project has been considered by the Regional Committee for Medical and Health Research Ethics (reference 2019/919) and has been ethically approved by the Norwegian Centre for Research Data (reference 924533).

## 4. Results

### 4.1. Selection of Studies

The systematic database searches retrieved 3835 records from CINAHL, Embase, Medline, PsycINFO, and Web of Science. After removing 1450 duplicates, 2385 records were screened by title, of which we found 2195 to be irrelevant. The remaining 190 articles were screened by abstracts and 92 of these were included for full-text reading. Full-text reading further excluded 29 studies (reasons for rejection: not original article, not immediate family members, participants not bereaved by suicide only, focus on events prior to suicide, etc.) and we ended up including 63 articles in this scoping review (see PRISMA diagram, Figure 1).

### 4.2. Characteristics of Included Studies

Figure 2 shows the key characteristics of the included studies. Of the 63 original articles included in the review, 24 studies were classified by method as quantitative [21,22,41,42,43,44,45,46,47,48,49,50,51,52,53,54,55,56,57,58,59,60,61,62,63], 29 as qualitative [14,15,64,65,66,67,68,69,70,71,72,73,74,75,76,77,78,79,80,81,82,83,84,85,86,87,88,89,90], and 10 as mixed-methods studies [49,91,92,93,94,95,96,97,98,99]. Almost ¼ were intervention studies [42,44,47,49,51,56,57,58,59,62,63,91,95,98,99]. The median sample size was 18 (min 4, max 250) for the qualitative studies, and most had 40 or less participants. One study using qualitative survey data had nearly 230 participants [79], and the other two studies with large sample sizes were qualitative analyses of observational data on the use of the internet [73,81]. The median sample size was 139 (min 8, max 11,572) for quantitative studies and 37 (min 5, max 270) for studies with a mixed design. The quantitative and mixed-method studies with less than 10 participants were intervention studies [42,99].

Forty percent of the studies were published in 2019 or later [14,21,22,45,46,47,48,51,55,56,58,60,64,65,67,70,79,80,82,83,87,89,92,94,98] and more than half of the studies were from Australia (*n* = 12) [14,22,41,47,59,61,67,72,74,76,80,95], the US (*n* = 12) [46,50,51,54,57,63,71,73,75,86,90,98] and the UK (*n* = 10) [52,53,64,65,68,69,78,79,89,93]. The remaining studies were from Ireland (*n* = 5) [66,82,88,97,99], Italy (*n* = 5) [21,55,56,70,87], Sweden (*n* = 5) [60,77,83,84,85], Canada (*n* = 4) [15,42,91,94], the Netherlands (*n* = 4) [43,44,49,81], Belgium (*n* = 1) [62], Denmark (*n* = 1) [45] and Lithuania (*n* = 1) [92]. Results from the network analysis of authors and citations of the included studies are presented in Appendix A and show that there are mainly clusters of authors publishing in this field. Within the 63 included articles, the most cited articles within the network [61,74] had about ten citations each. Some of the articles included in this review were not cited within the network but had several citations outside the network.

Few studies focused on specific survivor relationships; five on children [42,83,84,85,90], three on parents [75,82,89], one on siblings [77], and one on spouses [51]. Most studies involved bereaved adults (40), or all age groups or information on age were not provided (11). Only three comprised children [42,83,99], four adolescents [14,67,84,85], and five comprised ages 15 years or above (i.e., not children) [43,44,46,48,77].

### 4.3. Service/Support Types and Themes

The coding of service/support type resulted in nine categories (see Figure 3). These were: ‘service/support general’ (studies not focusing on a specific service or support type), ‘therapy’, ‘health services, other’, ‘support groups’, ‘postvention, other’, ‘online forums and chat’, ‘memorials’, ‘internet, general’ (studies taking a general approach to online/internet support), and finally ‘first responders’ (e.g., police, paramedics, funeral workers, coroner’s office). A few studies focused on a specific service organisation with several service/support types. These were coded as focusing on one service and were included in the “postvention, other” category.

Based on the content of the included studies we identified four main themes: (i) help-seeking and service use, which also captured barriers and facilitators for service/support resource use; (ii) needs and unmet needs for services/support; (iii) experiences and benefits, which captured experiences with the use of service/support resources and the perceived benefits of use (if, why and how); and (iv) effect and effectiveness, which comprised studies with a quantitative approach to study effect and impact. We found that themes differed among studies on different service/support types.

The largest service category, comprising 20 studies, included studies that did not focus on a specific service or support type but had a general focus. These were typically studies with a qualitative [14,15,67,74,75,76,77,78,79,80,88,90] or mixed [94,96,97] approach, and only five were quantitative studies [21,22,43,53,61]. Help-seeking and service use, needs and unmet needs, and experiences and benefits were common themes in the studies with a general focus, and few focused on effects.

Therapy, i.e., psychological treatment for mental and emotional problems, was the second largest service/support category. Ten studies fell within the therapy category; all but one [54] were intervention studies and all were either quantitative [42,44,51,54,57,58,62,63] or mixed [98,99]. All intervention studies focused on effects and comprised different modes of therapy. The last of the therapy studies was a survey on general experiences with individual therapy [54].

The third largest category, with nine studies, comprised postvention studies grouped together in a residual ‘postvention, other’ category. These were most often quantitative [45,47,55,56,59] or mixed [90,92], and two were qualitative [71,82]. Four of the articles were intervention studies focusing on effect [47,56,59,90].

The ‘health services, other’ category included five studies on health services other than therapy and specific postvention services. Two were qualitative [65,88], two were quantitative [46,52], and one was mixed [91]. Three studies concerned experiences with primary care/general practice [46,65,88]. Two studies focused on help-seeking and barriers for the use of specialised mental health care [52,91].

Six studies were on support groups [64,65,71,82,86,87], i.e., peer support in suicide bereavement groups moderated by a facilitator. In most studies the facilitator was a volunteer with survivor experience. However, this information was not always easily available. It was also not always stated whether the peer facilitator had received training. One study clearly stated that the group was facilitated by a clinician [86]. All six studies on support groups were qualitative and involved experiences with and benefits of this type of peer support.

Six studies were on online forums and chats. Of these, four were qualitative [70,81,84,85], one was quantitative [41], and one was mixed methods [49]. Five of six studies concerned online peer support forums [41,49,81,84,85]. Internet forums are online discussion sites, which typically are moderated. One study related to live chat support for people bereaved by suicide [70]. Live chat provides real-time conversations, and can serve as an alternative to, e.g., phone-based crisis and counselling services [ibid.].

Four studies related to memorials, i.e., initiatives that help to keep remembrance alive. Two of these studies were qualitative [68,73], one was quantitative [50], and one used mixed methods [95]. The latter study was a project organised by a suicide-prevention bereavement support service, and the other three were online memorials.

Two studies addressed suicide bereavement support via the internet in general. One was a qualitative study exploring several ways the internet is changing the experience of bereavement by suicide [69]. These included use of e-mail, social networking, and setting up website memorials. The other study related to a survey on the usage of digital resources by the suicide bereaved, with an additional focus on the use of online support groups and memorial websites [60].

In the category (non-healthcare) of first responders only one study was identified. This was a quantitative study on the impact of police behaviour during death notifications on the mental health of the bereaved [48].

Of the three studies with children as participants, two involved therapy [42,99] and one was on a family camp [83]. Of the four studies with adolescents as participants, two had a general focus (same first author) [14,67] and two focused on internet forums (same author) [84,85].

If we define postvention studies as those focusing on services or support types designed to target the suicide bereaved, all studies in the support group category, the memorial category, and ‘the postvention, other’ category were postventions. All but one of the studies in the therapy category were postventions; the non-intervention study [54] concerned experiences with therapy in general, i.e., services not specifically targeting the treatment of the suicide bereaved. Additionally, all but one study in the ‘online forums and chats’ category were postventions. The exception in this category was a study that also included an online forum not specifically targeting suicide survivors [85].

In addition to the ‘online forums and chats’ and ‘internet general’ category, the online service and support types identified were online memorials, internet-based therapy, and online booklets, as well as online support as part of other postvention services.

In our reporting of results, we present an extraction of relevant key findings with reference to the main themes involved (see Appendix A).

#### 4.3.1. Help-Seeking and Service Use

Results indicate that the bereaved seek help from and use a variety of service and support types [21,22,43,61,67,94]. This is also the case for bereaved adolescents; however, parental encouragement was found to be important in accessing adequate professional help [67]. Many started help-seeking in the first week and month after the death, but some did not find what they were looking for or found obstacles in help-seeking [21,22,67]. It is common to turn to the general practitioner (GP) for information, support and help [22].

Help-seeking behaviour is associated with different bereaved characteristics. A lack of social support may lead to the avoidance of reaching out to a psychologist or a GP, and result in looking for advice in online forums and relying on people outside of their immediate network, such as co-workers [21]. The physical and mental health consequences of grief (e.g., insomnia, poor appetite, low energy, increased anxiety, uncontrollable crying, anger, self-blame, depression, and guilt) may hinder help-seeking [74,96]. The results indicate that making initial contact with bereavement services can be particularly burdensome and people may be incapable of seeking help due to overwhelming grief responses [76,77]. Another factor found to reduce the likelihood of seeking professional help, as well as medication acceptance in some cases, was negative attitudes toward health services related to the deceased’s unmet needs [77].

Even though many bereaved use general health services, such as GPs, psychologists and counsellors, it can be difficult to find someone with experience with severe grief work or with specific training [21,61,74]. Furthermore, discontinuity, e.g., due to a high turnover of support staff, can deter further help-seeking [ibid]. Other barriers include a lack of information, a lack of awareness of services, a lack of available services, distance, and cost [14,21,61,79,88]. Participants from rural areas must often travel a long way to find support, which may be a barrier particularly for those experiencing health difficulties [74]. Need for specialised services for, e.g., LGBTIQA+, have also been voiced [88].

Results from a large survey among young adults found that people bereaved by suicide were less likely to have received informal support than those bereaved by sudden natural causes or unnatural causes, but did not differ from either comparison group on the receipt of formal support. However, the results suggested that people bereaved by suicide were less likely to have received immediate support and more likely to report the delayed receipt of support than people bereaved by sudden natural causes [53].

The ‘postvention, other’ category included two studies focusing on help-seeking. One was a study on the Danish Network for those Affected by Suicidal Behavior, an NGO that offers support, including counselling and participation in counsellor-led support groups to people affected by suicidal behaviour [45]. The study found that a third of users sought counselling within 6 months of the death. A help-seeking rate of 6 users per 10 suicide deaths was found within proximity to a counselling venue. Each additional 10 km of driving distance was associated with a 15% lower rate of use. Hence, geographical proximity to help centres was important for help-seeking and a barrier for access to support. The other study was an Italian study on the SOPROXI initiative, which provides information and offers support and treatment to the suicide bereaved in the form of brief psychotherapies, chat support groups, and residential mindfulness/self-compassion retreats [55,56]. The study found that the main reasons for contacting the service were seeking help for symptoms and connecting with other suicide survivors, while some did not know what to ask for. About one third of the participants participated in the “curing” intervention (psychotherapy, chat groups and meditation retreats). These participants were slightly older than the bereaved seeking the “caring” service (improving knowledge on suicide prevention, postvention and grief support), had typically survived a child or partner, and had more grief symptoms. They were also more likely to seek contact with other survivors compared to other bereaved seeking help from the service. Furthermore, they expressed hopelessness, loneliness and guilt more often than those who did not want to participate in the more structured intervention, despite reporting similar levels of depression [55].

#### 4.3.2. Needs and Unmet Needs

##### Needs in General

The bereaved typically have diverse support needs. Support needs are often related to the availability or absence of informal support by family or friends [97] and may differ within families [14]. Needs are subjective and relate to context of the suicide and the uniqueness of bereavement process, the individual personalities, individual coping experience, and developmental stages of the bereaved [88,90]. Most bereaved experience grief-related emotions, e.g., guilt and depressed mood, that occur so often and so strongly that they disturb everyday life [96]. Grief and health reactions co-occur; initial feelings of guilt, blame, shame, and anger are often manifested in enduring physical, psychological and psychosomatic difficulties [97]. A study of a long-term course of bereavement (8–10 years) found that suicide ideation was associated with an increased risk of long-term CG and depression. However, the risk of CG and depression was found to decrease over time [43].

Help to cope with grief responses such as extreme sadness, insomnia and nightmares, concentration difficulties, anger, self-blame, and fragmentary memories are common reasons for wanting professional help [77]. Those bereaved by suicide typically need help to process grief experiences, to cope with grief responses, learn skills and coping strategies, make meaning, understand and search for reasons leading to the suicide, tackle stigma, to normalize the grief experience, connect with other suicide survivors, to seek help, facilitate social relationships, self-reflect, remember, finding direction and move forward, as well as give back to society [14,15,75,77,88,97]. While some do not experience the need for help from health services [77,97], others may need help for many years [61].

These findings indicate that to meet the needs of individuals bereaved by suicide, service providers need to evaluate the bereaved persons’ coping strategies, and to be compassionate, open, and present in journeying with the bereaved following the pace of the individual to help bereaved individuals process their loss on an emotional level, and they may need to establish stability for those individuals in shock and provide psychoeducation on the grief experience [15]. Survivors also need to maintain control and agency in the relation to the services [14].

##### Specific Service Needs and Unmet Needs

Many bereaved experience unmet needs [14,78,94,97]. A survey study found that, overall, 94% of participants indicated a need for help to manage their grief but only 44% received help [61]. Individual unmet needs relate to information (first responders), medical/pharmacological needs (health services), support needs (health services, voluntary organisations), outreach (coroner’s office, health and social services, voluntary organisation), and collective unmet needs such as suicide pre-/postvention training and delivery (health and social services, first responders, voluntary organisations, and educational and professional associations) [14,94].

The aim of one of the studies was to identify areas of priority for improvements in suicide bereavement support in the UK [78]. The study pointed to three priority areas. A major gap in current service provision was found to be immediate outreach after suicide, a period when the bereaved lack the resources to identify and access support. First responders play a key role in filling this gap. Another priority was the diversification and development of peer support services (providing home visits, immediate outreach, email and telephone support, or access to a private message board). Additionally, the third was the provision of individual psychological support for those who feel suicidal. However, recognising suicide as a trauma is very important for survivors, and some form of individual trauma-based counselling was felt to be appropriate [78,88]. The study also illustrated service needs at the point of suicide discovery and later proactive support. First responders should provide immediate support: police must show compassion to avoid further stigmatising the death, and provide support and additional information on other support sources, while sudden death liaison workers should provide immediate emotional support, give information, and offer to connect with local bereaved peers. However, this immediate and proactive support should not be too intrusive. GPs should play a key role in later proactive support, including providing medical assistance if needed and screening and referring for appropriate treatment/support. The latter includes suicide support services offering peer support, face-to-face or group support, telephone helplines and email/web support services (bid.).

These findings are supported by other studies. Support needs changing during the course of bereavement; proactive and practical help is needed in the early stages since many do not know what they need in the immediate aftermath and they may be in need of practical help due to shock, numbness, and confusion [80]. A small sample study found that in nearly half of cases no initial offer of help was received; however, a high share of survivors would have liked to be contacted in the first couple of months after the suicide [94]. Furthermore, difficulties navigating services means that survivors need to be guided in what support is available and appropriate [80]. Experiences of stigma may lead to social isolation and the suicide bereaved may need help connecting with others with similar loss experiences (ibid.). Being proactively linked to ongoing formal support was perceived as a major unmet need [74]. Hence, findings indicate that the suicide bereaved need immediate and repeated contact, empathic and personal meetings, and information and grief-related support [77]. They need support and resources that are more flexible, accessible, and proactive [22]. Services need to be accessible on a long-term basis [14,88]. Short-term counselling may not be sufficient to deal with the complexity of the suicide [65]. Ethnic minorities may need help to tackle strong stigma and taboos related to suicide [79].

Contrary to what one might expect, a study of what makes assistance helpful for adolescents found that most study participants did not prefer online chat or helpline services; having a personal connection with and trust in the helper were more important [14]. However, reading online stories of real-life examples from peers of how they coped with the loss could be supportive for some. Adolescents were also found to strongly prefer help outside the family realm, since they then felt free to share things that they did not share with others and did not need to worry about upsetting their parents or other family members with their grief, or being met with pity [14]. Survivors may need support besides family and friends because they feel freer to share with others without having to worry about upsetting or wearing out family and friends by the need for talk, or because of fractured family relations [14,97].

#### 4.3.3. Experiences and Benefits

Studies show that survivors’ experiences with help and information from service providers and formal supporters, including first responders, ongoing support from professional and peer support groups, and internet-based resources varied greatly.

##### First Responders and Immediate Outreach

The responses of first responders and other professionals were found to influence the bereavement journey for suicide survivors [74], and the experience ranged from compassionate to cold (e.g., disrespectful and insensitive responses) [74,76]. Positive experiences were, e.g., sensitivity of the police in ensuring they were not left alone during the critical period immediately after the death, and offers to deliver the distressing news to other immediate relatives was perceived as particularly helpful [76]. The police could potentially have conflicting roles if the death was considered suspicious, which could lead to a cold approach [78]. Some studies found that police seldom provided information on the support alternatives, or the information was outdated or felt irrelevant [74,78]. The police’s competence in filling the role varied, and there was a perceived need to train first responders [ibid.]. In the immediate aftermath, unless the family/GP outreach teams take proactive measures, people that do not perceive they need help, or are unable to seek help, receive no support [78].

These results from studies of general experiences with services and support resources are echoed in the one (quantitative) study focusing on the impact of police behaviour during death notifications. The study found that a high share of the bereaved did not experience good support from the police, often because the police seemed overwhelmed with the situation themselves [48]. Some effect on mental health regarding the perceived behaviour of the police was found. The perception of police behaviour was closely related to whether or not the police left the bereaved before other services or family arrived. Furthermore, when the police handed out information material, the behaviour of the police was more positively rated.

The ‘postvention, other’ service category included a study from Australia [72], investigating how the Living Beyond Suicide program, a postvention service providing early support and which partners with crisis services such as the police and ambulance services, could be made more accessible and appropriate for Aboriginal people. The study found that Aboriginal participants had several suggestions on how to enhance the capacity of the service. One suggestion was to incorporate a process of listening and learning in the development of relationships and services, including Aboriginal capacity and people alongside experienced service providers. The composition of staff teams was noted to be of importance, with a need for the service to be staffed jointly by workers specialised in the kind of case management and advocacy required by the service, and workers connected to Aboriginal communities.

##### On-Going Support from Health Services

An Australian study from 2010 found that only 40% of those who received professional support felt satisfied with it [61]. Reasons for dissatisfaction with health services included lengthy access, insufficient care, a non-empathic encounter, perceived professional incompetency, and lack of appropriate training [61,76,77]. Reasons for satisfaction with health services, on the other hand, included receiving grief-related support, empathic encounters, and psychosocial benefits [77]. Relation to the helper, e.g., feeling connected, having trust, and feeling understood and validated is important for the experience of support [14,74,90].

Some experienced that GPs did not always provide appropriate response [78]. Of the three ‘health care, other’ investigating experiences with GPs, one was a survey which found that only half (48%) of the respondents reported that the responses from physicians were positive and help-offering, whilst 10% reported to have had negative responses. Respondent’s perceptions of physician responses were associated with differences in grief difficulties and mental health distress [46]. Two qualitative studies pointed to the potentially important role of GPs in the follow-up of the suicide bereaved [66,89]. They were found to be ideally positioned to take a proactive role (initiating contact), providing information, direction/signposting, and support during the grief journey. The studies found that this requires the GP’s awareness of suicide bereavement and the bereaved’s needs, including acknowledgement of the loss and lived life of the deceased, and the role of stigma and other barriers to help-seeking, including triage processes. Even though many suicide bereaved had positive experiences with their GP, GPs were often perceived as uncertain on how to respond to their loss [89].

The finding from general studies showed that some bereaved found it challenging to find clinical help from someone who was compassionate and experienced in grief and loss [74]. An Australian survey found that most people that had used face-to-face support perceived it as helpful [22]. Suicide support services may not be known, which delays access, and survivors also experience a shortage of supply with unwarranted geographical variations [78].

The survey on general experiences with individual therapy found that the participants had generally positive experiences [54]. However, a key finding was that many of those showing symptoms of PTSD did not report a formal diagnosis from a provider, which may lead to inadequate treatment of symptoms. Furthermore, closeness to the dead person and early intervention (entering therapy less than three months following the death) were factors found to be important for effective treatment. The findings also suggested that those who were most satisfied with therapy spent a longer time in therapy, but it was not clear if this was a dosage level effect or if their severity of symptoms required lengthier treatment.

Two studies focused specifically on experiences with mental health care. One study found that gaps in the healthcare system were the main barriers to seeking professional psychological help (e.g., consultation/therapy with a psychologist/psychotherapist/psychiatrist or a support group led by a professional psychologist) [92]. A characteristic of participants seeking help was reporting stigmatisation and feeling guilt; however, attitudes toward mental health specialists had the highest prognostic value in predicting help-seeking behaviours. The other study concerned the follow-up of bereaved families from psychiatric services experiencing patient suicide [52]. The study found that one of three relatives were not contacted after the death. Furthermore, a violent method of suicide was independently associated with a greater likelihood of contact with relatives. Four stigmatising patient-related factors were found to reduce the likelihood of services contacting relatives: forensic history, unemployment, and the primary diagnosis of alcohol or drug dependence or misuse.

The Swedish BRIS grief support program, a grief support camp (psychoeducational intervention) for families affected by a parent’s suicide organised by the non-profit organisation Children’s Rights in Society (Barnens rätt i samhället, BRIS) was recently evaluated [83]. The study investigated whether and, if so, how, children (*n* = 11) and parents (*n* = 11) perceived their participation in the program as meaningful. The study found that both children and parents highly valued their encounters with other suicide bereaved, leading to support exchange and normalisation, which again reduced “suicide stigma”, self-blame, and shame in their everyday life. The participants described a better-informed position in their grief, increased manageability, and better communication within the family. Parents reported an improved ability to support their children and had a more hopeful view of the future. Overall, the intervention was evaluated to have a relatively low cost compared to traditional approaches and was found to have great potential to dampen the negative effects of a suicide by helping families with psychological processing and de-stigmatization.

The ‘postvention, other’ service category also included a study evaluating Help is at Hand, a hardcopy and online booklet produced as part of England’s suicide-prevention strategy. It includes sections covering themes such as practical matters, experiencing bereavement (including coping strategies), bereaved people with particular needs, how friends and colleagues can help, and sources of support. An evaluation of the resource shows that it was largely well received [93]. The resource was extensively accessed online; however, copies were obtained more often by organisations than individuals and the main complaint was delay in gaining access. Use also required the promotion of resources.

One study of the ‘memorial’ category studies evaluated the satisfaction with the Lifekeeper Memory Quilt Project, implemented by the Salvation Army (SA), Hope for Life Suicide Prevention and Bereavement Support in Australia in 2008. The purpose of the project was both to provide support for those bereaved by suicide and to create greater public awareness of suicide [95]. The results of the evaluation showed high satisfaction with the project, and 92% of survey participants rated the Quilt project to be helpful or extremely helpful. Four themes were revealed: healing, creating opportunity for dialogue, reclaiming the real person, and raising public awareness. The project provided a milieu of understanding, acceptance, and connectedness, and a space to grieve free from stigma, judgment, and negative social reactions that encouraged bereaved people to discuss their loss from suicide. The results suggested that participation was perceived as being most helpful when it occurred more than one year post bereavement.

##### Peer Support: The Advantages of Support Groups

Findings suggest that peer support groups for individuals bereaved by suicide can be effective for individuals who have a need for an external outlet for their grief and may reduce demand on healthcare systems if they are adapted to individual needs [64]. Support groups are described to be complementary to one-to-one intensive support such as counselling and therapy [65,71].

Support groups are found to fill a range of grief-related needs, including a place for continuing to bond and memorize, e.g., [82]; meaning making and making sense of the death, e.g., [65,82,86,87]; normalising their grief experiences, e.g., [65,86]; finding a place for blame and taming the tendency to self-blame, e.g., [82,87]; providing a safe space for authentic self-expression and expressing their innermost feelings, including disappointing interactions they had with other people, without the fear of upsetting others or feeling judged, e.g., [64,65,82,86].

Suicide bereavement is often characterised by a feeling of social isolation, due to stigmatization and self-stigmatization. Peer support groups provide an arena for a sense of belonging, finding companionship, mutual understanding, empathy and acceptance, as well as sharing ways of coping with the death, e.g., [64,65,82,86,87]. These experiences help to counteract social isolation and stigma and normalize the grief experience [74,76,88]. The support group as a safe space or sanctuary seems to be a key feature, enabling the bereaved to connect with others with similar experiences in a way in which others who have not been bereaved by suicide cannot offer. Support groups also provide a place to belong and to reframe social relationships. Sharing a common trauma leads to the collective transformation of perceptions and views of suicide, creating a collective sense of identity which feels empowering, e.g., [65,87]. The role of the support group also appears to be crucial for the process of finding hope, rebuilding the self, and recognizing one’s own growth, e.g., [82,86].

Contextual factors and mechanisms affecting the grief processes and seeking peer support were found to include, e.g., the cultural stigma of suicide; self-stigma; gender norms (‘toxic masculinity’); not wanting to burden loved ones due to judgement and a lack of understanding; difficulties adjusting to the suicide; and practical challenges of accessing groups, e.g., [64,65]. Support group affiliation has been found to be a time-limited activity for most of those bereaved by suicide [71].

##### Possible Challenges with Support Groups

Not all experiences with support groups are positive. Some of the suicide bereaved find support groups unproductive, and some find it difficult and upsetting to listen to others retelling their stories, leading to possible re-traumatisation [65,74,80]. For some bereaved, having the same type of kinship or nature of relationship with the deceased is crucial for the experience of the support group to be beneficial. Other factors contributing to negative experiences which could make people leave the group include facilitator/leadership deficiencies; unbalanced participation (more needy individuals that persistently take over meetings); changing composition of the support group; informal social relations and shared norms between members leading to some perceiving the support group as “cliqueish” [65,71].

##### Facilitator Skill Issues

A reflection expressed by some users was that non-professional support groups did not provide healing strategies or alternative approaches to coping, and often lacked organization and direction. This could be improved by the education and training of peer facilitators on coping strategies and processes of group facilitation or by using professional facilitation [74,80]. Some regarded a trained facilitator as best; however, regular supervision for volunteer peer support workers was felt to represent a more economical option than professional facilitation [78]. One study in particular pointed to the importance of group facilitator skills. Good leadership was characterised by ability to prevent monopolization, to be attentive to nonverbal communication and gently encourage group participation, and (sometimes) keeping silent and letting other members respond first. The authors expressed that most peer survivors need training to develop leadership skills [71].

##### One-to-One Peer Support

Even though many suicide bereaved value peer support in groups, some would prefer one-to-one peer support [74]. The Peer Support Program, a one-to-one peer support program introduced by the Canadian Mental Health Association (CMHA) Suicide Services in Calgary was evaluated in a study focusing on the experiences and benefits from the viewpoint of both peer supporters and their clients [91]. The peer supporters considered the helpful aspects of the meetings to be providing instrumental strategies for coping and considering family concerns. Suggesting resources for the clients also gave the peer supporters a feeling of being useful and able to help. The benefits expressed by the clients were at large the same as for support groups. Talking about extended family matters was a particularly supportive aspect of the meetings.

##### Use of Internet Resources

A study of internet use found that participants used sites to inform others about the death, making sense of the events and gaining support from an internet community of others who had been similarly bereaved. Few participants mentioned negative experiences; however, a few participants preferred not to use the internet or did not have access to a computer [69]. The authors concluded that the internet transformed the bereavement experiences, giving access to other people’s experiences. Others have found that use of internet can be both helpful and disturbing [80].

Another study of internet use, which focused on the use of online support groups and memorial websites, found that that a majority used digital resources in their grief work [60]. However, gender as well as socio-demographic differences in use of services were found, i.e., women more often used support groups, but not memorial sites, and the users of the memory sites had lower levels of education than non-users. Furthermore, time since the suicide was associated with the use of online support groups, but not memory sites. Losing a child was associated with a higher intensity of use, but not the propensity to use digital resources. The psychosocial ill health among the suicide bereaved participants was severe, but severity did not predict the propensity to engage in online support groups or memorial websites. However, higher online support group activity predicted greater satisfaction with current psychosocial health, which may indicate that it is an effective way of coping with grief related to suicide loss. Memorial websites, on the other hand, have been found to have the opposite effect, which could suggest that memorial websites may increase rumination.

The results also suggest that the respondents are using digital resources independently of how they value other forms of help [60]. Hence, the use of digital resources seems to be a supplement rather than an alternative to the use of other sources of support, such as psychiatric or primary care. One survey on the use of internet forums found that most participants had sought face-to-face help from sources other than internet forums, e.g., mental health professionals (70.3%), face-to-face support groups (45.5%), and professional help online (e.g., e-counselling or accessing information from the website of a suicide bereavement organization (9.8%)). Another study of users of internet forums found that the most common health service support was having visited a health professional such as a psychologist/psychiatrist. It was not so common to have visited the GP related to the loss [49]. An Australian survey found that face-to-face support was judged as very helpful more often than online resources, which more often was judged as somewhat helpful [22].

##### Internet Forums

Internet forums were found to be perceived as beneficial for the majority of users [41,49], and the forum was valued for finding recognition and for its anonymity, low threshold, and open-hearted atmosphere [49]. While few limitations with use of internet forums were found in one study [41], criticisms of the forum were also expressed among the people interviewed in the other study [49], including giving a depressive feel, being cluttered and needing more structure, low reaction speed to messages sometimes, too few messages posted, messages being too old to react to, and too little positive news.

A study on what the bereaved communicate on the forums found that a range of self-help mechanisms were used, such as sharing personal experiences, often with emotional expressions of grief with the most apparent content in messages, expressions of support or empathy, providing advice, and universality (recognition). Messages including experiences with healthcare services appeared only occasionally [81].

The potential role of internet forums to help renegotiate suicide stigma and break the silence surrounding parental suicide for young suicide survivors was the theme of two Swedish studies [84,85]. The forums helped to give a voice to the parentally bereaved, to unburden the emotional pain, and to construct meaning in the parent’s suicide ((re)interpretation of the suicide as a selfish act, or as the fault of the child). Three types of stigma-related dynamics on the responses of others were identified. Two related to the reassurance of ‘normalcy’, either through the dichotomy between ‘perpetrator’ and ‘victim (typically by the non-suicide bereaved), or through the sharing of similar experiences. The third mechanism related to reinforced ‘othering’, which may appear in the cases where contact with other suicide-bereaved youths is not achieved. The results illustrated the important role of the website editors in narrative framing for the interpretation and organization of the suicide experience (‘involuntary suicide’, ‘failing society’), which worked as a resistance to the ‘suicide stigma’. Additionally, the power of accumulated experience can potentially work to destigmatize and empower parentally bereaved participants’ grief.

##### Live Chat

In the one study on live chats, the benefits were presented in five linked themes [70]: meaning-making, reactions to the loss, resources, needs (practical support, emotional support, advice), and interactions with the operator (exploration, support, reconstruction). The live chat served as a safe space allowing for the disclosure of non-socially desirable details and to help make sense of suicide through the reconstruction of events and the deceased’s motivations. Reasons to resort to live chats for support and reassurance included limited social resources and dissatisfaction with the available formal support. The study concluded that because of their anonymity and accessibility, live chats represent a valid first-line form of support, from which survivors may obtain useful information and start a meaning-making process.

##### Online Memorials

One study of 250 memorials from two memorialization websites found that the majority of the memorials were posted by family members of the deceased and were mostly (80%) written for young males. They typically had the format of a letter or an obituary. Analyses of the memorials identified fourteen themes, ranging from sadness and expressions of love and other grief reactions of the bereaved, to acknowledgment of the suicide and the search for reasons for the death. The researcher concluded that online memorialization seems to fill two functions: continuing bonds with the deceased and strengthening bonds with the living [73].

One study compared the written memorials on websites posted by survivors of suicide with those written by people who had lost a significant other from natural causes using a computer program. Significant differences were found. Memorials written by survivors of suicide had longer sentences and used longer words, had more death-related words, had fewer references to the self or to the deceased (“you”), and more words reflective of anger and sadness. The author concluded that the results suggest the deaths from suicide had a more profound impact on the survivors than the natural deaths and resulted in greater emotional distress [50].

A study interviewing owners of suicide memorial sites [68] found that the use of websites and Facebook pages for online memorials fulfilled a range of functions spanning across social, emotional, and practical realms. Online memorialisation enabled the bereaved to connect with others with similar experiences and enabled a form of uncensored self-expression. Online memorialization assists the grieving process because it allows for the exteriorisation of feeling. Memorials can help alleviate trauma by the process of making and managing a new cultural script; by discovering new aspects of the deceased’s personality through postings by others who have known them; and because the memorial site or Facebook page becomes a mourning object. The results also disclosed problematic issues that suicide survivors must tackle when engaging with online memorials: tailing off interest due to time lapse, feeling responsible for the thoughts and actions of others, censorship, unwanted interest from strangers, and changes to the site without their consent. A point of particular concern was the removal of the page without their consent, which is experienced as a double loss that can serve to compound the grief. The authors conclude that online memorials transform the experiences of the bereaved and open up new ways of grieving and managing trauma, however, not without pitfalls potentially exacerbating the grieving process.

#### 4.3.4. Effects

##### Health Services and Peer Support

A German survey found that unmet need for professional support, regardless of whether or not they had sought support, was associated with reported increased levels of sorrow, lack of energy, and guilt [96]. Bereaved who reported receiving enough professional support still reported an increased occurrence of several negative emotions, but not of feelings of guilt, sorrow, or anger about the deceased. These are associations and not causal relationships.

A cohort study of a long-term course of bereavement (8–10 years) showed no effect of cognitive behavioural therapy, support from the GP, and/or mental health care, while mutual support was associated with an increased risk of CG [43].

##### Therapy

Two intervention studies were on individual therapy [51,63]. One showed positive effects of CG therapy on post-traumatic stress disorder (PTSD) and CG among widowed survivors of veteran suicide [51], and one found that CG therapy was an acceptable and promising treatment for suicide-bereaved individuals with CG, but raised concern about the acceptability of medication alone as a treatment for CG in suicide-bereaved adults [63]. Three studies were on group therapy [42,57,99], of which two concerned children [42,99] and one concerned bereaved adults with CG [57], all reporting positive results of group therapy interventions. One of the therapy interventions was on family-based cognitive behaviour grief therapy and showed a reduction in the risk of maladaptive grief reactions among suicide ideators [44]. Cognitive behaviour therapy (CBT)-based psycho educational intervention with home visits showed no significant effect on the development of CG reactions, depression, or suicide risk factors among suicide survivors. However, the intervention showed a reduction in other grief reactions, and may therefore serve as supportive counselling for suicide survivors [61]. A mixed methods study evaluating the Artful Grief Studio for military suicide survivors showed significant improvements in social validation, while the qualitative findings also found benefits and positive experiences [98]. An internet-based cognitive behavioural therapy study for the bereaved suffering with prolonged grief disorder (PGD) concluded that this represents an effective treatment approach, and an alternative to face-to-face grief interventions [58].

##### Community-Based Postvention Services

The ‘postvention, other’ category included two effect studies of StandBy, a community-based suicide bereavement service in Australia that provides support and a coordinated response for people bereaved by suicide [47,59]. Both studies showed that the programme was effective in reducing adverse outcomes associated with suicide bereavement and may have a positive effect on other health and social outcomes.

##### Voluntary Resources Providing Support and Treatment

Another ‘postvention, other’ study investigated the effect of the mindfulness retreat offered by the Italian SOPROXI initiative, and found a significant reduction in most dimensions of patient-reported profiles of mood states, and lower levels of overidentification were observed after the retreat [56].

##### Internet Forums

A study followed visitors of two online support forums for persons bereaved by suicide over 1 year and examined the mental health changes [49]. Participants were mostly female, low in well-being, with high levels of depressive symptoms and CG. Some positive changes after one year were observed, but many were still struggling with their mental health.

## 5. Discussion

### 5.1. Summary of Results

Of the 63 included articles in this scoping review, 62% used qualitative or mixed methods covering a range of different health services and support types. Most of the studies on support groups and internet-based resources (online forums, chats, memorials, etc.) were qualitative and focused on the bereaved’ s experiences with the services. The articles applying quantitative methods (38%) mainly comprised effect studies on therapy interventions, postvention services, or general services/support studied with surveys or registry data. There was a boost in publications in this field from 2019 onwards; 40% of the studies between January 2010 and March 2022 were published in 2019 or later. It is likely that extra focus on mental health and the risk of suicide during the COVID-19-pandemic has contributed to the steep increase in publications.

#### 5.1.1. Geographical Distribution

More than half of the 63 studies were performed in Australia, the US and the UK, and there is generally a lack of studies on this topic from several European countries (the exceptions are Sweden, Italy, and Ireland, with at least five publications each since 2010). Andriessen (2014) reviewed the literature on suicide bereavement and postvention in major suicidology journals and concluded that the understanding of suicide bereavement and the provision of survivor support might benefit from studies from other parts of the world than Western countries [10]. As health service researchers we also see a general need for the publications in this field to contextualise, explain, and discuss the national health service and support system settings in which postventions and other services for those bereaved by suicide are provided.

#### 5.1.2. Survivor Relationship and Demographic Characteristics

Only 10 of the 63 included studies focused on specific survivor relationships, half of them on losing a child, and the rest on losing a parent, sibling or spouse. Given the results in Ali and Lucock (2016), who found that many suicide bereaved prefer to have peer support from someone who has experienced the same type of loss, future studies should aim to make a separation between different types of survivor relationships [65]. A review by Pitman et al. (2014) elaborates on the role of kinship for the effects of suicide bereavement on mental health and suicide risk [9].

More than two thirds of the included studies had adult participants, while only eight studies focused on children or adolescents, implying that the voice of the young bereaved is underrepresented in the literature. Gender balance in the studies is also lacking. Due to the overrepresentation of adult men committing suicide, there will be more bereaved women than men, and women tend to participate more than men in studies on suicide bereavement [22,45,100]. Erlangsen et al. (2021) found that women in Denmark bereaved by suicide were less likely to be hospitalised for somatic disorders and see general practitioners, compared to the general population [45]. Westerlund et al. (2020) found that women more often used support groups, but not memorial sites [60]. These gender differences in study participation and help-seeking behaviour means that we must be careful when interpreting the findings. More knowledge on the follow-up, support, and experiences of men bereaved by suicide is needed.

Few of the included studies examined how different ethnic minority groups may have various experiences with suicide bereavement, and we need future research to distinguish between different subgroups of the suicide-bereaved population. This finding was also noted in a previous review by Maple et al. (2018) [24]. The Goodwin-Smith et al. (2014) paper on how to develop a culturally appropriate and accessible suicide postvention service for Aboriginal communities in South Australia suggested a systematic joint combination of staff with Aboriginal background and case management specialists to accommodate Aboriginal needs and respect Aboriginal culture. Rivart et al. (2021) claimed to be the first study in the UK to investigate the experiences and support needs of individuals bereaved by suicide from ethnic minority backgrounds. They found that ethnic minority groups experienced a need to tackle stigma-related issues within the ethnic minority groups, and that they reported a lack of support despite their attempts to engage with services [79].

#### 5.1.3. The Importance of Peer Support

When we first planned this scoping review, our intention was to focus on formal health services only. However, preliminary searches and studies of former reviews clearly showed that peer support as a source of help is of particular importance for the suicide bereaved, probably more so than for those bereaved by other causes of death [9,50]. It is not always a clear-cut line between “professional” and “non-professional” peer support resources, or the degree of “health service” in a peer support resource. Some peer support resources are initiated by health services and led by health professionals [92], while others are driven by NGOs or user organisations with trained facilitators or counsellors [45,71]. Often, the origin, funding, and organisation of the peer support resource is not well described in research articles, although this can be an important contextual factor. To gain more knowledge on whether formal health services should take a more active and systematic approach in including peer support as part of their “service menu”, researchers need to describe the context more thoroughly and do research on effects (for both participants and facilitators) of different ways of organising and staffing peer support resources. Services and resources for the suicide bereaved are likely to benefit from a more systematic and quality-focused approach.

The existing literature on the suicide bereaved’s positive experiences with peer support shows that this is doubtlessly an important supplement to formal health services for this group. Peer support users have reported benefits that counteract social isolation and stigma and normalize the grief experience that are not necessarily found in formal health services. Examples of such features are finding a safe space for emotional expression and sharing, finding companionship, mutual understanding, a sense of belonging, and feeling validated [74,76,89]. Since the evidence on support groups experiences is mainly based on qualitative data, it would be useful for the field to see more quantitative studies on the effect of support groups [78]. For countries with access to high-quality individual-level registry data, such as the Nordic countries, longitudinal population-based observational studies with linkages between health registries can be performed, given that support groups register each participants’ personal ID number and start/stop-date for participation. A national registry on peer support participation might be a good idea for facilitating future research in this field.

We found that few articles on peer support commented upon the initiation and facilitation of the peer support resource, nor the skills of the counsellors/facilitators, which is a limitation of these studies. Feigelman and Feigelman (2011) pointed to the importance of group facilitator skills, emphasizing the ability to prevent monopolization, to be attentive to nonverbal communication, and gently encourage group participation without dominating too much themselves [71]. A systematic review on the effect of support group peer facilitator training programmes on facilitator and group member outcomes for people with medical illnesses ended up including only one RCT which concluded that *“Well-designed and well-conducted, adequately powered trials of peer support group facilitator training programmes for patients with medical illnesses are needed”* [101]. Our findings suggest that the same is true regarding peer support for people bereaved by suicide.

Feigelman and Feigelman (2011) and Ali and Lucock (2016) presented possible challenges with support groups: hearing others’ stories or sharing their experiences can be upsetting (leading to possible re-traumatisation); unbalanced participation (more needy individuals that persistently take over meetings); changing the composition of the support group; and informal social relations and shared norms between members leading to some perceiving the support group as “cliqueish”, have been mentioned as possible negative experiences with support groups [65,71]. Overall, it is important to recognise the subjectivity of needs in the suicide bereaved, which means that peer support should be diversified, providing not only peer support groups but also one-to-one peer support, home visits, etc. [78].

#### 5.1.4. Health Services and Postventions

There is a general lack of studies on suicide-bereaved patient trajectories in the health services and on the association between health service demand and supply for this group. Longitudinal studies are needed to provide evidence on health service use at different time points before and after the suicide, and on outcomes regarding both somatic and mental health, as well as working life outcomes such as sick leave, vocational rehabilitation, unemployment, and disability pension for suicide survivors. Studies of follow-up for the suicide bereaved from general community mental health and addiction services were lacking, while studies on postvention services were more common [45,47,55,56,59,62,91,93], although the evidence of effectiveness was found to be low in a systematic review by Andriessen et al. (2019) [36].

Findings from the study of counsellor-led support groups (postvention) in Denmark showed that each additional 10 km driving distance gave 15% lower user rates [45]. Hence, many bereaved do not have a postvention service nearby, and must use general health services. This is an argument for increasing the general knowledge about suicide bereavement in local health services. It also implies positive expectations for the effect of future internet-based postvention services that can operate regardless of geography.

We found several studies on psychological treatment [42,44,45,51,54,57,58,62,63,98,99], with the most frequently studied diagnosis being complicated grief (CG) [43,49,51,54,57,62,63]. However, there are a lack of studies on other mental disorders, somatic diagnoses, and comorbidity. Ohye et al. (2020) studied the intensive outpatient treatment of PTSD and CG in suicide-bereaved military widows and concluded that many of their respondents had PTSD symptoms, but no diagnosis, which again may lead to the inadequate treatment of symptoms [51]. This confirms the findings of underdiagnosed PTSD in the suicide bereaved found in Sandford et al. (2016), who argue that therapists might need training in the specifics of how suicide grief is different [54]. Again, we argue that education in the specifics of suicide grief is something that should yield all health services. Furthermore, treatment in health services might cause harmful effects [102], and the potential downsides of treatment and support could be more explored in future research on suicide bereavement.

The general practitioner is often pointed to as having an ideal position for the follow-up of those bereaved by suicide [66,89]. However, they seldom have the tools to initiate or complete a proactive follow-up of their patients, so the responsibility of making contact falls on the patient. Additionally, the bereaved report a lack of understanding and compassion from GPs and other (non-postvention) health services [46,66,89]. Similar experiences have been found regarding first responders. In Hofman et al. (2021), the police were found to be overwhelmed by the situation themselves and unable to respond adequately according to the bereaved [48]. A study that explored GP’s experiences with dealing with parents bereaved by suicide showed that GPs had low confidence in dealing with suicide and that they felt unprepared to face the parents. Some of them felt guilt surrounding the suicide, which again made them reluctant to initiate contact with their patients [103].

#### 5.1.5. Internet-Based Resources

A recent systematic review by Lestienne et al. (2021) examined the use and benefits of online resources dedicated to people bereaved by suicide [27]. They found moderate quality evidence from 12 articles (most of which also included our scoping review) for several positive effects of online resources. One advantage of digital resources is that they provide help-seeking opportunities 24/7 for less educated, more disadvantaged, and isolated people. Frequent users were middle-aged women, parents who had lost their child, and recently bereaved individuals. Future interventions should use this knowledge to tailor the services and reach out to other groups as well. Although negative effects of digital resources were rare, it was noted that website editors or online facilitators have an important role in moderating and framing the interpretation and organisation of the suicide experience. With the pandemic nudging more people onto digital platforms, we expect to see an increase in interactive digital support groups for people bereaved by suicide. Digital platforms provide opportunities for mixing suicide bereaved across geographical regions, making it possible to focus support and services on the same type of survivorship.

### 5.2. Implications for Practice and Future Research

Having scoped the literature on health service use, follow-up, and support for people bereaved by suicide, we see several important implications for practice. There is a need for all of us to acknowledge that suicide bereavement is a special kind of bereavement that can be extra hard to cope with. Keywords are feelings of stigma; taboo; self-blame; anger; or even relief. The probability of meeting someone bereaved by suicide who needs empathy and support is high for most over the life course. It could probably make a difference if the average level of knowledge on grief in general, and what is special about suicide bereavement, increased in the whole population—through education and prevention strategies at schools, colleges, universities, and in population campaigns [104].

The suicide bereaved are likely to meet first responders (e.g., police, paramedics, funeral workers, and the coroner’s office) in the immediate aftermath of the suicide. This first meeting with these services can be of great importance for future help-seeking behaviour and service use. However, the literature shows that the suicide bereaved’s experiences with first responders are not always good. A lack of empathy and genuine understanding, a lack of information, insecurity, and service providers being overwhelmed themselves have been pointed to as key challenges in first responder services. Moreover, the bereaved are often in shock, making communication and understanding of the information given difficult. Ensuring a continuum of care for the bereaved is an important task for first responders and health professionals. While remembering to continuously obtain consent from the bereaved during the patient pathway, one health service should not leave before contact with the next is made. Not everyone feels that they need any help, but many of the bereaved report that they were not offered help in the right place at the right time.

We know from the literature that people bereaved by suicide seek help from a variety of service and support types, many starting the first weeks and month after the death typically with primary care emergency teams, their GP, and internet-based resources. As described above, several GPs do not feel competent enough in consultations with people bereaved by suicide. Knowledge on suicide bereavement should be given systematically as part of all medical training, especially for GPs. Given that GPs are systematically contacted by first responders (with consent from the patient), GPs should contact bereaved patients proactively and follow them up over time, as the bereaved may hesitate to contact help, or might not see the need for help while still in shock. One of the most severe gaps in the literature is knowledge of the timing of health service use and how to ensure a systematic follow-up of the bereaved. Hence, future research should prioritize longitudinal studies of health service use and support trajectories of the suicide bereaved, including longitudinal registry studies and cohort studies, such as the one presented for adults by de Groot and Kollen (2013) [43]. National health authorities should provide national recommendations and minimum standards for the level of skills and knowledge health professionals and peer support counsellors/facilitators should have about suicide bereavement. Furthermore, we need a deeper understanding of how and why specific subgroups of the population (e.g., children, men, ethnic minority groups, religious groups, LGBTIQA+, etc.) may have special needs in health and support services that differ from the rest of the population.

### 5.3. Strengths and Limitations

This scoping review takes a broad approach in scoping and presenting results from the existing literature on the support, follow-up, and health service utilisation of families bereaved by suicide, which is rare in the literature. A general limitation of scoping reviews is the lack of critical appraisal of the included studies, meaning that we are not able to identify gaps in the literature based on quality and bias issues. The literature on suicide bereavement includes a variety of methods and designs, with a relatively large proportion of qualitative and mixed-methods research that is challenging to undertake quality assessment on. Hence, we found the scoping review methodology [30] with a broad perspective on health service use and support suitable for our study rationale, which was to provide the basis for the further development of research questions and practice recommendations within a comprehensive research project.

We may have missed some relevant studies due to the choice of databases searched and the time constraint (January 2010–March 2022). However, we have tried to compensate by running extra hand-searches in CoCites and systematically going through references in existing reviews. The limitation of immediate family was set because we assumed that the follow-up of bereaved in the health services after a suicide most often is directed towards the immediate family. We acknowledge that this is a limitation of the search, as other relatives, friends, and colleagues can be defined as bereaved as well [8], but we considered experiences with systematic follow-up in the health services and participation in support groups mainly to yield immediate family members [61].

The scoping review only included English literature from high-income countries, and we acknowledge that literature in other languages and middle- and low-income countries may provide other evidence. We expect our restrictions to increase the probability of comparable levels of health services and support in the literature included.

## 6. Conclusions

This scoping review has shown that we have some knowledge about the need for, and experiences with, health services and support resources for people bereaved by suicide, but a lack of knowledge about their help-seeking behaviour, patient pathways, coordination between services, and long-term outcomes. Suicide bereavement requires an extra dimension of compassion and understanding from service providers and peer supporters. Based on the evidence included in this review, there is a need for raising competence on how to meet people bereaved by suicide and how to ensure a continuum of care in all stages of the bereavement trajectory among all types of first responders and health professionals. The supply of health services and peer support resources must be diversified and adapted to the individual needs of people going through this trauma. Evidence on internet-based resources for this group shows promising results, with the opportunity to provide treatment and support regardless of geographical boundaries. It is important that health professionals and peer supporters are adequately trained, and the potential harmful effects of treatment and service use for this group should be debated further in future studies. To know for sure what effects services and support for this group have, we need more longitudinal observational studies of health service use and patient trajectories for people bereaved by suicide.

## Figures and Tables

**Figure 1 ijerph-19-10016-f001:**
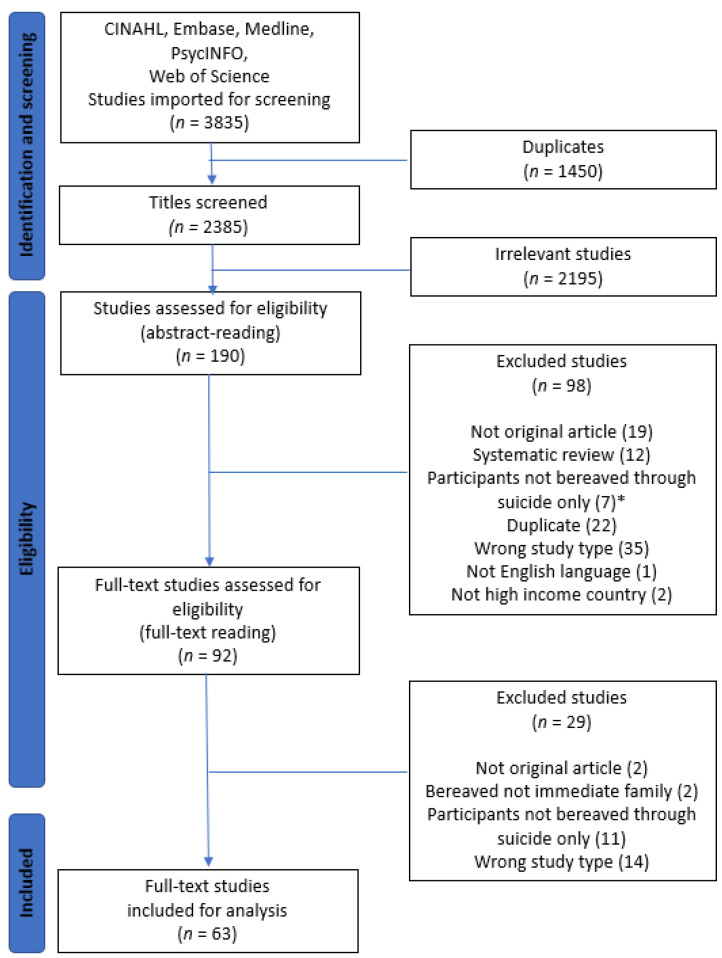
Search results—PRISMA diagram: Preferred Reporting Items for Systematic Reviews and Meta-Analyses. * Excluded if the results were not presented for the suicide bereaved separately.

**Figure 2 ijerph-19-10016-f002:**
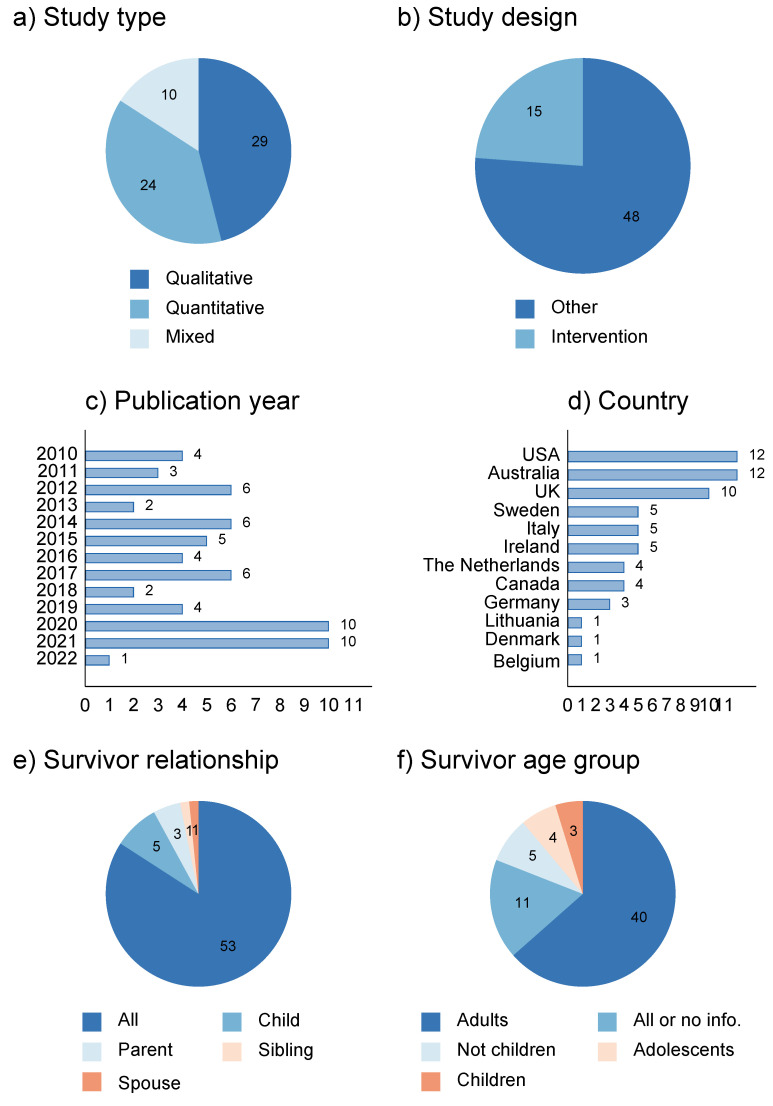
Study characteristics for study type, study design, year of publication, countries (based on affiliation of first author), survivor relationship to the deceased, survivor age group. Number of studies. *n* = 63.

**Figure 3 ijerph-19-10016-f003:**
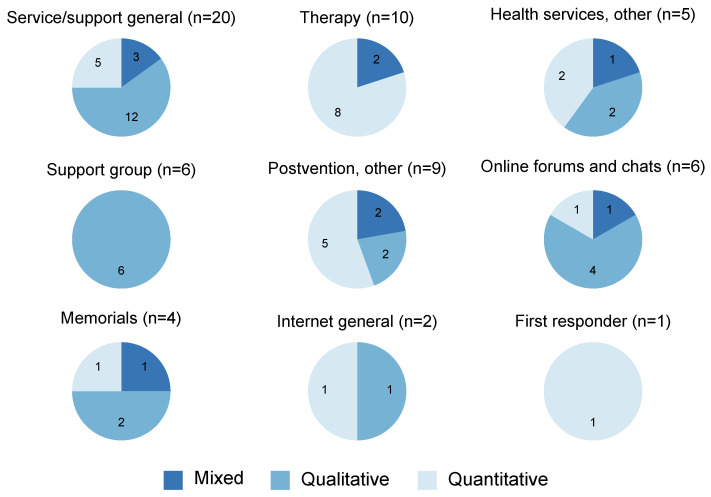
Service and support type. Number of studies. Distribution on study type. *n* = 63.

**Table 1 ijerph-19-10016-t001:** Inclusion and exclusion criteria.

**P—Population**	Suicide bereaved: immediate family (parents, spouse/partner, siblings, and children of any age).
**C—Concept**	Different trajectories in the health services and health-care utilisation (such as primary care, specialised care, outpatient care, extramural care), support groups and other support resources (other than close personal social network).
**C—Context**	Language limited to English. Studies in developed countries defined as high-income countries according to the World Bank (2020). Publication period in the past 10 years from January 2010 to March 2022.

**Table 2 ijerph-19-10016-t002:** Search strategy for concepts: (a) suicide, (b) bereavement and (c) health services. Asterisk (*) used for truncation.

**(a) Suicide**	**Suicide** (**MeSH**)
	suicid*
**(b) Bereavement**	Bereavement (esh)
	bereave* OR grie* OR mourn*
**(c) Health services/support types**	Health Services (MeSH) OR Delivery of Health Care (MeSH) OR Primary Health Care (MeSH) OR Secondary Care (MeSH) OR Community Health Services (MeSH) OR Mental Health Care (MeSH) OR Outpatient (MeSH) OR Inpatient (MeSH) OR Hospitals (MeSH) OR Ambulatory Care Facilities (MeSH) OR Ambulatory Care (MeSH) OR Counseling (MeSH) OR Referral and consultation (MeSH)
	“health care” OR healthcare OR health-care OR inpatient OR outpatient OR treat* OR support* OR follow-up OR “follow up” OR counsel* OR consult* OR hospital* OR ambulatory* OR team* OR postvention

## Data Availability

Not applicable.

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
