# Peer review of "Use of Health Services and Support Resources by Immediate Family Members Bereaved by Suicide: A Scoping Review"

_ijerph, 2022, doi:10.3390/ijerph191610016_

Round 1

Reviewer 1 Report

This is a vey lengthy review of the literature on the use of services for suicidal bereavement. A Scoping Review protocol on this topic was published in 2020.  It is a comprehensive review of the literature.

The Lestienne et al., (2021) systematic review is discussed in several places and it feels like a review of previous reviews.

The manuscript requires editing for English grammar and spelling throughout. Maybe a careful editing would improve readability.  In its present form, it feels dense and difficult to get through.  The subheadings make it easier to read.

The methodology appears sound.

The paragraphs on the role of first responders is of interest since this would be the first place to intervene in providing support to a survivor.

It seems strange that there were no studies on the use of grief counselors within the counseling and therapy sphere.  Did any of the papers look specifically at grief counseling?

Suicide rates had been increasing before COVID.  COVID did raise concerns about increases in suicide, but there were many other factors that were more important than COVID that resulted in higher rates in the U.S. and Australia.

Author Response

Please find the author's reply to the review report in the attached file. 

Reviewer 2 Report

International Journal of Environmental Research and Public Health (IJERPH)

Review Letter to Authors

Manuscript Title:           Use of health services and support resources by                                                      immediate family members bereaved by suicide. A                                                scoping review.

Date:                              Saturday, July 30, 2022

_________________________________________________________

This scoping review included studies from 2010 to March 2022 that investigated the follow-up and support in health services, peer support services and other resources available (e.g., internet-based resources) of families bereaved by suicide. The authors followed the scoping review framework provided by the Johanna Briggs Institute and performed a double-blinded screening process using Covidence.

The authors provided a work that was informative, well written, and provides a great foundation for future research. The methodological rigor was impressive and the figures in the manuscript were very detailed and provided excellent clarity. I enjoyed reading this manuscript, and recommend the authors carefully read the manuscript and correct any spelling and/or grammatical errors. It is in the spirit of improving the manuscript that I offer the following comments and/or questions:

On Page 10, you wrote:

One study were found on live-chat support for people bereaved by suicide [71].

Change to:

One study related to live-chat support for people bereaved by suicide [71].

On Page 10, you wrote:

Four studies regard memorials, i.e., initiatives that helps to keep remembrance alive.

Change to:

Four studies related to memorials, i.e., initiatives that helps to keep remembrance alive.

On Page 11, you wrote:

The other study was survey on the usage of digital resources by suicide bereaved, with an additional focus on use of online support groups and memorial websites [61].

Change to:

The other study related to a survey on the usage of digital resources by suicide bereaved, with an additional focus on use of online support groups and memorial websites [61].

On Page 13, you wrote:

The aim of one of the studies was to identify areas of priority for improvements in suicide bereavement support in UK [79].

Change to:

The aim of one of the studies was to identify areas of priority for improvements in suicide bereavement support in the UK [79].

On Page 13, you wrote:

The study also provided illustration of service needs at the point of suicide discovery and later proactive support.

Change to:

The study also provided an illustration of service needs at the point of suicide discovery and later proactive support.

OR

The study also illustrated service needs at the point of suicide discovery and later proactive support.

On Page 14, you wrote:

These findings are supported by other studies. Support needs change during the

course of bereavement; proactive and practical help is need in the early stages since many do not know what they need in the immediate aftermath and they may be in need of practical help due to shock, numbness and confusion [81].

Change to:

These findings are supported by other studies. Support needs change during the

course of bereavement; proactive and practical help is needed in the early stages since many do not know what they need in the immediate aftermath and they may be in need of practical help due to shock, numbness and confusion [81].

On Page 14, you wrote:

They need support and resources that are more flexible and accessible, and are offered more proactive [22]. Services need to be accessible on a long-term basis [14,89]. Short-term counselling may not be sufficient to deal with the complexity of the suicide [66]. Ethnic minorities may need help tackle strong stigma and taboo around suicide within certain ethnic minority groups [80].

Change to:

They need support and resources that are more flexible, accessible, and proactive [22]. Services need to be accessible on a long-term basis [14,89]. Short-term counselling may not be sufficient to deal with the complexity of the suicide [66]. Ethnic minorities may need help to tackle strong stigma and taboo related to suicide [80].

On Page 14, you wrote:

Contrary to what one might expect, a study of what makes help helpful for adolescents found that most study participants did not prefer online chat or helpline services, relations, having a personal connection with and trust in the helper, were more important [14]. However, reading online stories of real-life examples from peers of how they coped with the loss, could be supportive for some. Adolescents were also found to strongly prefer help outside the realm family, since they then felt free to share things that they did not share with others and did not need to worry about upsetting their parents or other family members with their grief or being met with pity [14]. Survivors may need support besides family and friends because they feel freer to share with others without having to worry about upsetting or wearing out family and friend by the need for talk or because of fractured family relations [14,98].

Change to:

Contrary to what one might expect, a study of what makes assistance helpful for adolescents revealed most study participants did not prefer online chat or helpline services, relations, having a personal connection with and trust in the helper was most important [14]. However, reading online stories of real-life examples from peers of how they coped with the loss could be supportive for some. Adolescents also strongly preferred receiving help from outside the family because they felt free to share things that they did not share with others and did not need to worry about upsetting their parents or other family members with their grief or being pitied [14]. Survivors may need support besides family and friends because these forms of support may allow them to feel freer to share with others without having to worry about upsetting or emotionally draining family and friends due to the need to talk or fractured family relations [14,98].

On Page 15 you wrote:

In the immediate aftermath, people that is unable to seek help or do not perceive that they need help do not get any support unless proactive measures are taken by the family/GP/outreach teams [79].

Change to:

In the immediate aftermath, unless the family/GP outreach teams take proactive measures, people that do not perceive they need help or are unable to seek help, receive no support [79].

On Page 15, you wrote:

The 'postvention, other' service category included a study from Australia [73], investigating how the Living Beyond Suicide program, a postvention service providing early support and which partners with crisis services such as the police and ambulance services, could be made more accessible and appropriate for Aboriginal people. The study found that Aboriginal people valued the kind of interventions and support provided by the service.

* What was it about this intervention that appealed to the Aboriginal people?

On Page 15, you wrote:

Reasons for dissatisfaction with health services include lengthy access, insufficient care, nonempathic encounter, and perceived professional incompetency and lack of appropriate training [62,77,78].

Change to:

Reasons for dissatisfaction with health services include lengthy access, insufficient care, a non-empathic encounter, and perceived professional incompetency and lack of appropriate training [62,77,78].

On Page 22, you wrote:

Only ten of the 63 included studies focused on specific survivor relationships, half of them on losing a child, the rest on losing a parent, sibling og spouse.

Change to:

Only ten of the 63 included studies focused on specific survivor relationships, half of them on losing a child, the rest on losing a parent, sibling or spouse.

On Page 22, you wrote:

Given the results in Ali & Lucock (2016), who found that many suicide bereaved prefer to have peer support from someone who have experiences the same type of loss, future studies should aim to be able to separate between different types of survivor relationships [66].

Change to:

Given the results in Ali and Lucock (2016), who found that many suicide bereaved prefer to have peer support from someone who have experiences the same type of loss, future studies should aim to make a separation between different types of survivor relationships [66].

On Page 23, you wrote:

Feigelman & Feigelman (2011) pointed to the importance of group facilitator skills, emphasizing the ability to prevent monopolization, to be attentive to nonverbal communication and gently encourage group participation without dominating too much themselves [72].

Change to:

Feigelman and Feigelman (2011) pointed to the importance of group facilitator skills, emphasizing the ability to prevent monopolization, to be attentive to nonverbal communication and gently encourage group participation without dominating too much themselves [72].

On Page 25, you wrote:

The suicide bereaved is likely to meet first responders (e.g., police, paramedics, funeral workers, coroner's office) in the immediate aftermath of the suicide.

Change to:

The suicide bereaved is likely to meet first responders (e.g., police, paramedics, funeral workers, and coroner’s office) in the immediate aftermath of the suicide.

On Page 25, you wrote:

The probability of meeting someone bereaved by suicide who needs your empathy and support is hight for all of us over the life course.

Change to:

The probability of meeting someone bereaved by suicide who needs empathy and support is high for most over the life course.

Recommendation:

In your “Strengths and Limitations” section, I recommend that you acknowledge the few empirical works that have examined the various experiences of those bereaved by suicide, particularly as it relates to race and ethnicity. [I mention this because of the question that I posed regarding the intervention that appealed to the Aboriginal people that you mentioned on Page 15 of your manuscript. [I copied my question below]

On Page 15, you wrote:

The 'postvention, other' service category included a study from Australia [73], investigating how the Living Beyond Suicide program, a postvention service providing early support and which partners with crisis services such as the police and ambulance services, could be made more accessible and appropriate for Aboriginal people. The study found that Aboriginal people valued the kind of interventions and support provided by the service.

* What was it about this intervention that appealed to the Aboriginal people?

Author Response

(The authors gave the same response as above.)
